# Cohort Profile: The Xiamen Pubertal Growth Cohort Longitudinal Study

Xijie Wang [1,2,3], Yanhui Li [3], Di Gao [3], Zhaogeng Yang [3], Bin Dong [3], Yanhui Dong [3,*], Zhiyong Zou [3,*] and Jun Ma [3]

1   Vanke School of Public Health, Tsinghua University, Beijing 100084, China
2   Institute for Healthy China, Tsinghua University, Beijing 100084, China
3   Institute of Child and Adolescent Health & School of Public Health, Peking University, No. 38, Xueyuan Road, Haidian District, Beijing 100191, China
*   Correspondence: dongyanhui@bjmu.edu.cn (Y.D.); harveyzou2002@bjmu.edu.cn (Z.Z.); Tel.: +86-10-82801624 (Y.D. and Z.Z.); Fax: +86-10-82801178

**Abstract:** This cohort was set up to capture pubertal onset among children in China and to analyze how timing of pubertal onset would influence cardiovascular disease risk in later life. Pubertal onset was defined as secondary sexual characteristics (SSC) attained Tanner II stage, which was breast development for girls and testis of 4 mL for boys. Meanwhile, height growth spurt, defined as age of take-off and age of peak height velocity, were also involved to observe the consistency between the three indicators in discriminating pubertal onset. The study was conducted in Xiamen, China from November 2017 to November 2020 with 6-month gaps. One thousand, four hundred and sixteen children from four project schools who had not yet started puberty were involved at baseline. By November 2020, 1272 children were still under follow-up, with 945 (74.3%) of them reaching Tanner II stage. We would continue to follow the pubertal development, as well as change in crucial risk factors for cardiovascular disease in these participants. Evidence from the present cohort study would help to reveal the influence of pubertal growth on long-term cardiovascular health and would be one of the very first studies to provide such evidence from Asian countries.

**Keywords:** cohort profile; pubertal development; growth and development; blood pressure; school-aged children





## 1. Introduction

Puberty is one of the most crucial stages in human growth and development [1,2] and is accompanied by a series of complex biological landmarks under the cooperation of the adrenal gland and the gonadal system, including height growth spurt, development of secondary sexual characteristics, voice maturation, body hair growth and even mood changes [3,4]. Much of the literature has documented that the age of puberty has declined significantly over the past decades [5,6], which increases significant concern about children entering puberty at younger ages across the world [7]. Previous studies reported that early puberty onset could lead to various adverse outcomes, such as adolescent deviant behaviors, short stature, obesity, increased risks of chronic diseases, higher blood pressure and other metabolic disorders in later life [8,9]. Additionally, delayed puberty was also reported to have relationships with adverse events, such as impaired fertility and compromised bone health [10].

In some children, puberty might take place prematurely to produce early puberty onset, whilst, in others, it fails to be switched on at the appropriate time, leading to delayed puberty. Thus, puberty timing, featured by early, delayed or on time, refers to an individual's status as a referent group or a set of norms. The age at each milestone could be used to represent one's onset of puberty and could be used to determine one's timing of pubertal onset according to relative status among children of the same sex [11,12].

Among all the indicators used for determining individual pubertal onset, secondary sexual characteristics of Tanner stage II is widely considered as the golden standard to evaluate one's pubertal onset timing [11,13,14]. However, as the secondary sexual characteristics require frequent physical assessment, other indicators including age at height take off (ATO) and age at peak height velocity (APHV) were also used in recent studies to evaluate individual timing of pubertal onset [15,16].

Nutritional status, socioeconomic status, adoption, geographical migration and emotional well-being all had effects on the pubertal timing in children, and the identification of pubertal timing phases during adolescence also varied by study design and the specific indicator used to reflect pubertal onset [17]. For instance, results from The Northern Finland Birth Cohort 1966 found early APHV was an independent risk factor of metabolic disorders in both male and female [18], while another study in the United States found that self-reported pubertal timing had no significant relationship to late metabolic health status in males [19]. Therefore, it was important to determine a proper practical indicator of pubertal onset when analyzing the impact of pubertal timing on later health outcomes and to have it verified in children of different nutritional status, since body fat is a widely accepted influence factor of pubertal timing [20]. However, the correlation of ages determined by different indicators and their consistency for evaluating one's pubertal timing were seldom confirmed by longitudinal studies. Thus, it was necessary to compare the applicability of multiple indicators of pubertal timing evaluation in children.

To observe the true status of pubertal growth and to analyze how it may be associated with long-term cardiovascular health, a population-based, observational, longitudinal cohort study was designed and conducted in Xiamen, China, with support from the National Natural Science Foundation of China. The present manuscript describes the initial cohort profile, the objectives and the implementation of the Xiamen Pubertal Growth Cohort Longitudinal Study.

## 2. Materials and Methods

### 2.1. Study Participants

To maintain the stability and sufficiency of sample size, all four municipal nine-year education schools in Xiamen city of China were involved in the cohort study. The preliminary research was conducted based on data from the sixth round of the Chinese National Survey on Students Constitution and Health (CNSSCH) and native school physical examinations. Data from the sixth round of CHSSCH found that the median age of spermarche and menarche were 14.0 and 12.3, respectively [21,22], while previous research and local practice both showed that the majority of children's secondary sexual characteristics did not appear until the age of 10 (boys) and 9 (girls). To capture the exact time of pubertal onset (represented by appearance of secondary sexual characteristics), cluster sampling was used in this study, and all the boys from grade 3 and 4 (age of 8 to 9) and girls from grade 2 and grade 3 (age of 7 to 8) were invited to participate regardless of their biological sex and ethics. A total of 2125 students without significant organ diseases were invited to participate. In November 2017, 1416 out of 2125 (66.6%) children whose parents signed the informed consent and whose secondary sexual characteristics had not developed were recruited to the cohort. Binary biological sex, categorized as boys and girls, was used for all sex-specific examinations in the present cohort.

### 2.2. Ethical Consideration

The study was approved by the Institutional Review Board of Peking University (No. IRB 00001052-17026). Prior to participant recruitment, we held a parent meeting with students and their parents from targeted grade levels in each of the project schools. The meetings were held by the research team and school nurses and introduced the objectives, specific contents and implementation methods of the cohort. Students and their parents were given adequate chance and time to ask questions. Thereafter, written informed consent forms were given to students and parents. They were told to bring the informed consent

forms back home, to fully discuss it with the student and family members and to decide whether they would take part into this study. Those who agreed to participate in the study submitted an informed consent form signed by both student and the parent the next day.

*2.3. Physical Examination*

From November 2017 to November 2020, follow-ups were conducted every 6 months (except for May 2020 when it was canceled due to COVID-19). Physical examination (including the Tanner stage of pubertal development, weight, height, waist circumference and body composition) and blood pressure was measured in each wave of follow-up, while blood biochemical and sex hormone measurements, child and parent questionnaires were collected each year. The timetable of follow-ups, measurements, correspondent devices and methods are displayed in Table 1.

The Tanner stage of pubertal development and pubic hair was measured with four fixed physicians (2 males and 2 females). The child was asked to take off all their clothes (except for underpants) in a warm and completely enclosed room, and the specific Tanner stage of secondary sexual characteristics and pubic hair were determined by two physicians of the same sex together, based on the Tanner stage pictures and testicular development models [23]. When the actual stage was between 2 stages, the less developed stage was recorded; when the stages of breasts or testis on two sides were different, the more developed stage was recorded [24]. Pubertal onset was determined as girls' breast and boys testis volume attaining Tanner stage II and was recorded as Tanner-II age. The pictures of Tanner stage and testis models are displayed in Supplementary Figure S1.

Weight, height and waist circumference were measured by experienced physicians of the same sex in warm and closed rooms. Children were asked to wear undergarments and be barefoot for these measurements. Weight was measured twice by an electronic weight scale (Tanita HD-394, TANITA, Tokyo, Japan) to the nearest 0.1 kg, with a difference between two measurements less than 0.5 kg. Height was measured by a portable stadiometer to the nearest 0.1 cm, with a difference between two measurement less than 0.5 cm. Body mass index (BMI) was calculated as weight (kg)/height$^2$ (m$^2$), while age and sex specific BMI z scores were converted on the basis of the growth reference released by the World Health Organization in 2007 [25]. Waist circumference was measured twice by a body measuring tape (Myotape, Accufitness, LLC., Denver, CO, USA) to the nearest 0.1 cm, located at the midline of the participant's armpit and at the midpoint between the lower part of the last rib and the top of the hip. The difference between the two measurements should be less than 0.5 cm. The final measurement of weight and height, as well as the smaller measurement of waist circumference, were recorded. Body composition was measured with the bioelectrical impedance method (Tanita MC-180, TANITA, Tokyo, Japan). Children were asked to wear light clothes, be barefoot and remain silent during measurement.

Systolic blood pressure (SBP) was determined by onset of the first Korotkoff sound, and diastolic blood pressure (DBP) was determined by the fifth Korotkoff sound. Blood pressure was measured twice with a 5-min gap from the right arm, following the 2017 version of clinical practice guideline for screening and management of high blood pressure in children and adolescents, which was published by the American Academy of Pediatrics [26]. If the difference between two measurements $\geq$ 10 mmHg (either SBP or DBP), extra measurements would be conducted until the difference between the last two measurements was less than 10 mmHg. For the baseline conducted in November 2017, blood pressure was measured with a mercury sphygmomanometer (XJ11D, Shanghai Medical Instruments Co., Ltd., Shanghai, China). For the following follow-ups, it was measured with an electronic monitor (Omeron HBP-1100, Omron Corporation, Kyoto, Japan). To test the consistency of the two measurements, 200 children were randomly selected in May 2018, each child was measured twice with two measurements, and the consistency between two mercury measurements and two electronic measurements, as well as between the mercury and electronic measurements, was analyzed and displayed in Supplementary Figure S2. Basically, the difference in mercury and electronic measurements on SBP and DBP were both within 10 mmHg, which was within the acceptable range of implementation requirements.

**Table 1.** Timetable and measurements for cohort follow-up.

| Time / Measurement | November 2017 | May 2018 | November 2018 | May 2019 | November 2019 | November 2020 | Device | Method |
|---|---|---|---|---|---|---|---|---|
| Sample size | 2125 | 1416 | 1372 | 1370 | 1338 | 1272 | - | - |
| Weight | ○ | ○ | ○ | ○ | ○ | ○ | Electronic weight scale (Tanita HD-394, TANITA, Tokyo, Japan) | Barefoot, to the nearest of 0.1 kg, measured twice |
| Height | ○ | ○ | ○ | ○ | ○ | ○ | Portable stadiometer (model TZG, Bengbu, China) | Barefoot, to the nearest of 0.1 cm, measured twice |
| Waist circumference | ○ | ○ | ○ | ○ | ○ | ○ | Body measuring tape (Myotape, Accufitness, LLC., Denver, CO, USA) | Without shirts, to the nearest of 0.1 cm located at midline of the participant's armpit, at the midpoint between the lower part of the last rib and the top of the hip, measured twice |
| Tanner stage | ○ | ○ | ○ | ○ | ○ | ○ | Tanner stage atlas and testis development model | Physical examination measured with 2 fixed staff |
| Body composition | ○ | ○ | ○ | ○ | ○ | ○ | Tanita MC-180, TANITA, Tokyo, Japan | With light clothes, barefoot, in silence |
| Blood pressure | ○ | ○ | ○ | ○ | ○ | ○ | Mercury sphygmomanometer for November 2017 (XJ11D, Shanghai Medical Instruments Co., Ltd., Shanghai, China) Electronic monitor for follow-up (Omron HBP-1100, Omron Corporation, Kyoto, Japan) | With proper cuff, in silence, measured twice, with a difference within 10 mmHg |
| Blood biochemical | ○ | | ○ | | ○ | ○ | Automatic biochemistry analyzer (Roche Cobas, Roche, Rotkreuz, Switzerland) | After 12 h overnight fast. Sample was centrifuged at 3000r for 10 min, aliquoted and stored at −80 °C |
| Urine phthalates | | ○ | | | | | Standards solutions (Toronto Research Chemicals Inc., Toronto, ON, Canada) | Morning urine sample, stored at −80 °C |
| Serum sex hormone | ○ | | ○ | | ○ | ○ | Radioimmunoassay system, (XH6080, Xi'an Nuclear Instrument Factory, Xi'an, China) | |
| Students' questionnaire | ○ | | ○ | | ○ | ○ | Background information, habits on dietary intake, habits on physical activities, knowledge on healthy lifestyle | |
| Guardians' questionnaire | ○ | □ | ○ | □ | ○ | ○ | Child's birth factors, family background information, parental health status, parental habits on dietary intake, parental habits on physical activities, parental knowledge on healthy lifestyle, family history of noncommunicable chronic diseases. | |

○ indicated the measurement was conducted during the follow-up, while blank indicated that the measurement was not conducted at the follow-up.

The accuracy of the above instruments was verified in previous pediatric scientific research [27–30], and all the instruments were calibrated according to the manuals and workbooks before the daily examination. All examinations requiring the removal of outer clothing were performed in a completely closed room, and the project team would make sure that the doors, windows and curtains were closed before the start of each day's examination. During the examination, only one entrance and exit to the room was reserved and was covered by a screen to ensure that students' privacy was protected. The examinations including height, weight and Tanner stage required removing clothes and were, therefore, conducted in private and closed rooms and by physicians with the same biological sex. This process was approved by the ethics committee and was strictly followed throughout the study.

*2.4. Sample Collection*

Blood samples were taken by qualified nurses in the morning from 8:00 to 10:00. Children were asked by school nurses and teachers to stop eating after 8 o'clock the night before the blood draw. Approximately 3 mL blood was taken from each child for fasting plasma glucose, lipid profile, estradiol and testosterone tests. The blood sample was stored in the $-4\ °C$ refrigerator immediately, and be transported to the laboratory for centrifugation (3000r for 10 min) and sub-packaged as soon as the physical examination ended each morning. All the blood samples were stored in a $-80\ °C$ refrigerator before and after they were transported with cold chain to Beijing. All the samples were examined by a qualified bioassay company at the same time, after all the required samples were collected. The blood biochemical examination was performed with the automatic biochemistry analyzer (Roche Cobas, Roche, Rotkreuz, Switzerland); estradiol and testosterone were analyzed by a radioimmunoassay system (XH6080, Xi'an Nuclear Instrument Factory, Xi'an, China).

A urine sample was collected by healthcare workers at the first wave of the visit. All urine specimens were stored at $-80\ °C$ for examination. Seven standard solutions of phthalates metabolites, including mono-methyl phthalate, mono-ethyl phthalate, mono-n-butyl phthalate, mono-iso-butyl phthalate, mono-2-ethylhexyl phthalate, mono-2-ethyl-5-oxohexyl phthalate and mono-2-ethyl-5-hydroxyhexyl phthalate, were purchased from Toronto Research Chemicals Inc. (Toronto, ON, Canada) to test the PAE metabolites in urine samples by the Peking University Medical and Health Analysis Center.

The children and parents' questionnaires were developed based on the information, motivation and behavioral skills model and was used for practice in a multicentered intervention study [31]. Children were asked to fill in the questionnaire in class, with team members reading and giving necessary instructions or explanations. The parents' questionnaires were filled in by one of the child's guardians. Four major subheadings were included in the children's questionnaire, which were: background information, habits on dietary intake, habits on physical activities and knowledge on healthy lifestyle. Seven major headings were included in parents' questionnaires, which were: child's birth factors, family background information, parental health status, parental habits on dietary intake, parental habits on physical activities, parental knowledge on healthy lifestyle and family history of noncommunicable chronic diseases.

*2.5. Quality Control*

A rigid quality control process was conducted in the whole procedure of this cohort and were addressed through the following practice. Prior to subject recruitment, the present study went through scientific and ethical review by the National Natural Science Foundation of China and Institutional Review Board of Peking University. Implementation of the cohort strictly followed the reviewed form of study protocol. During data collection, the research team provided technical guidance and participated in both data collection and quality control procedures. On each day of data collection, 5% of subjects were randomly selected for a quality control recheck, all examinations were to be repeated if the recheck failed to satisfy the implementation protocol (which did not happen in practice). For data

handling, all data entry was conducted by two separate stuff to reduce processing error; all biological samples were strictly preserved according to the requirements and transported to a laboratory with testing qualification in Beijing for uniform testing after the completion of all field work.

## 3. Key Findings

### 3.1. Primary Results

Three indicators were used to discriminate the age of pubertal onset for each child, which were Tanner II age, ATO and APHV. ATO and APHV were calculated with the Preece and Baines Growth Model 1 (PBGM 1) [15,16] based on height records and predicted adult height for each child. The results of PBGM 1 was found to have good agreement with the actual age of ATO and APHV, and it was, therefore, commonly used for calculation of ATO and APHV in epidemiology studies [32]. First, adult height was calculated with the following function developed by Cole T.J.: $height_{adult} = SD_{adult} \times (z_{child} \times r) + M_{adult}$, where r was the coefficient between the individual child's height and adult height [33]. Second, the ATO and APHV for each child were calculated with PBGM1 with the "pbreg" package in Stata. The population-based mean and SD of adult height in urban Fujian was obtained from the 2014 Chinese National Survey on Students Constitution and Health [34]. All the parameters are displayed in detail in Supplementary Table S1.

Nine hundred and forty-five children (53.3% boys) had attained Tanner II stage by November 2020 and were involved in the calculation of Tanner-II age. The mean ages of included and excluded children were 8.6 (SD: 0.8) and 8.4 (SD: 0.7), respectively. The included children were also slightly taller and heavier than the excluded children at baseline. The difference in age for the height spurt was significant in boys, with an ATO of 8.2 (SD: 1.6) for included and 8.5 (SD: 1.5) for excluded children and APHV of 11.4 (SD: 1.1) for included and 11.1 (SD: 1.5) for excluded children. The difference in girls was insignificant. Tanner II age was 11.0 (SD: 0.8) and 9.7 (SD: 0.9) for included boys and girls, respectively. The demographic characteristics at baseline and children's age of height growth are displayed in Table 2.

**Table 2.** Demographic characteristics at baseline and their age of pubertal growth.

| Variables | Included | Excluded | *p* Value |
|---|---|---|---|
| All | | | |
| Sample size, n (%) | 945 | 327 | |
| Boys, n (%) | 504 (53.3) | 163 (49.8) | 0.117 |
| Age, year, mean (SD) | 8.6 (0.8) | 8.4 (0.7) | <0.001 |
| Height, cm, mean (SD) | 133.1 (7.2) | 131.4 (7.1) | <0.001 |
| weight, kg, mean (SD) | 29.8 (8.0) | 28.3 (7.4) | 0.002 |
| BMI, kg/m$^2$, mean (SD) | 16.6 (3.0) | 16.2 (2.9) | 0.017 |
| Boys | | | |
| ATO, year, mean (SD) | 8.2 (1.6) | 8.5 (1.5) | 0.065 |
| APHV, year, mean (SD) | 11.4 (1.1) | 11.1 (1.5) | 0.007 |
| Tanner II age, year, mean (SD) | 11.0 (0.8) | - | - |
| Girls | | | |
| ATO, year, mean (SD) | 7.4 (1.0) | 7.3 (0.9) | 0.265 |
| APHV, year, mean (SD) | 9.3 (1.2) | 9.1 (1.2) | 0.113 |
| Tanner II age, year, mean (SD) | 9.7 (0.9) | - | - |

SD, standard deviation; ATO, age of height take-off; APHV, age of peak height velocity.

### 3.2. Discussion

To our knowledge, the Xiamen Pubertal Growth Cohort Longitudinal Study is one of the very first longitudinal studies tracking pubertal onset by Tanner stage. Findings from this cohort provide valuable real-world information for pubertal growth and development patterns in Chinese children, revealing its role and potential mechanisms in influencing blood pressure health. The most significant strength of the present cohort was the start and

frequency of follow-up settings. Through the intensive follow-ups, which started before pubertal onset, we were able to depict the timing and order of the two major pubertal events in Chinese children for the first time. Although we were unable to make horizontal contrast as the existing research data were quite limited, the order of major events from the present study was similar to what Tanner and Marshall found in British children from the last century, while the timing was overall earlier [13,14]. Although the regular follow-ups were disturbed by the COVID-19 pandemic, the initial results of this cohort already provided information on the timing of pubertal growth and development in Chinese children. We will continue to follow this cohort every one to two years to explore the association and potential mechanisms between pubertal growth and long-term cardiovascular health. We will mainly focus on cardiovascular health outcomes, including weight, blood lipids, blood glucose during adolescence and in adulthood, clinical cardiovascular events, as well as their association with genetic and environment circumstances.

However, there are limitations with the present cohort. First, as all the participants grew up in Xiamen city and went to fixed nine-year schools, the socio- and natural environment of participants was highly consistent. Therefore, the Tanner II age reported from the present analysis could not represent the whole population of China. Extrapolation of the conclusions was, therefore, limited and should be performed with caution. Second, ATO and APHV were fitted with models instead of actual observation. Although we predicted adult height for each individual to obtain more precise calculations for model fitting, there could be differences with their actual age of height spurt.

## 4. Conclusions

The Xiamen Pubertal Growth Cohort Longitudinal Study is one of the first longitudinal studies focusing on pubertal growth and its relationship to cardiometabolic health in Chinese children. The initial results of this cohort provided information on the timing of pubertal growth and development in Chinese children, and we will continue follow-up to explore the association and potential mechanisms between pubertal growth and blood pressure health.

**Supplementary Materials:** The following supporting information can be downloaded at: https://www.mdpi.com/article/10.3390/future1010003/s1. Supplementary Figure S1: Tanner stages of secondary sexual characteristics and pubic hair for male and female. Supplementary Figure S2: Bland-Altman plots plot of systolic blood pressure and diastolic blood pressure difference measured by mercury sphygmomanometer and electronic monitor. Supplementary Table S1: Population-based height in urban area, Fujian, and r-values used for prediction of adult height.

**Author Contributions:** Conceptualization: J.M. Data curation: X.W., Y.L., D.G., Z.Y. Formal analysis: X.W., J.M., Z.Z. Funding acquisition: Y.D., B.D., J.M. Methodology: X.W., Y.D., B.D., J.M. Project ministration: X.W., Y.D., B.D., J.M. Visualization: none. Writing—original draft: X.W. Writing—review and editing: Y.D., B.D., Z.Z., J.M. All authors have read and agreed to the published version of the manuscript.

**Funding:** This work was supported by the National Natural Science Foundation (grant number 81673192 awarded to Jun Ma, 81773454 and 82073573 awarded to Zhiyong Zou, 82103865 awarded to Yanhui Dong and 82204067 to Xijie Wang), and the China Postdoctoral Science Foundation (National Postdoctoral Program for Innovative Talent, BX20200019 and 2020M680266 to Yanhui Dong).

**Institutional Review Board Statement:** The study has been approved by the Institutional Review Board of Peking University (No. IRB 00001052-17026). All the children and their parents signed the informed consent and agreed to take part in the research.

**Informed Consent Statement:** Informed consent was obtained from all subjects involved in the study.

**Data Availability Statement:** The cohort data are not freely available, but our team welcomes collaborations with other researchers. For further information, please contact the corresponding authors.

**Acknowledgments:** The authors greatly appreciated the Educational Administration Leaderships and primary and middle school health nurses who worked hard on data collection.

**Conflicts of Interest:** The authors declare no conflict of interest.

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
