# Peer review of "Cohort Profile: The Xiamen Pubertal Growth Cohort Longitudinal Study"

_future, doi:10.3390/future1010003_

Round 1

Reviewer 1 Report

Comments and Suggestions for the authors include the following:

The authors might consider including details and rationale in relation to the      statistical analysis for the study (e.g. model, statistical method).

There was not any information concerning the reliability and validity of instrument and measure for study utilization.

Although the study setting is in China, study inclusion and exclusion criteria do not clearly address participants' ethnicity. If applicable, could the authors address ethnicity in relation to study participant inclusion and exclusion criteria?

The authors might also consider the use of conceptual and operational definitions for Tanner stage II with respect to gender (e.g female girls, male boys).

Author Response

The authors might consider including details and rationale in relation to the statistical analysis for the study (e.g. model, statistical method). 

There was not any information concerning the reliability and validity of instrument and measure for study utilization.

--Thank you very much for the comments. We have provided more details and references on the measurement tools used in the study, the statistical analysis methods applied and the choice of models (line 164-166; line 226-228).

Although the study setting is in China, study inclusion and exclusion criteria do not clearly address participants' ethnicity. If applicable, could the authors address ethnicity in relation to study participant inclusion and exclusion criteria?

--Thank you very much for the comments. Subject recruitment for this study was conducted in Xiamen, China, and all subjects were of Chinese nationality, with the majority being Han Chinese students and only a very few ethnic minority students. Because the subjects all lived in Xiamen city and grew up in similar environment, we did not specifically restrict the subjects' ethnicity. This is also added at the inclusion exclusion criteria (line 80-84; line 258-260).

The authors might also consider the use of conceptual and operational definitions for Tanner stage II with respect to gender (e.g female girls, male boys).

--Thank you very much for the comments. We considered the binary biological sex for Tanner stage categories and all other sex-specific indicators, and have added this explanation to method section (line 86-88).

Reviewer 2 Report

The manuscript should be accepted for the publication and requires minor-revision according to the comments to the authors.

The article written on English language has 7 pages 2 tables and 27 references and consists of sections: Abstract, Introduction, Materials and methods, Key findings, discussion, conclusion and references. The article presents the protocol of the longitudinal research conducted on 1416 prepubertal children and will contribute to the knowledge of the influence of pubertal growth on long-term cardiovascular health.

The Abstract reflects accurately the contents of the manuscript.

The Introduction states the problem adequately and reviews background adequately for the protocol study.

In the Materials and Methods section, participants are described adequately. In description of methods criteria of assessing weight status as normal weight, overweight and obese should be stated, as well as of the blood pressure.  In description of sample collection, examined biochemical parameters should be stated.

In the Key findings, a part of results is presented adequately for the protocol study. The strengths and weaknesses of the study are well described.

The number and choice of references is satisfactory.

Author Response

The manuscript should be accepted for the publication and requires minor-revision according to the comments to the authors.

The article written on English language has 7 pages 2 tables and 27 references and consists of sections: Abstract, Introduction, Materials and methods, Key findings, discussion, conclusion and references. The article presents the protocol of the longitudinal research conducted on 1416 prepubertal children and will contribute to the knowledge of the influence of pubertal growth on long-term cardiovascular health.

The Abstract reflects accurately the contents of the manuscript.

The Introduction states the problem adequately and reviews background adequately for the protocol study.

In the Materials and Methods section, participants are described adequately. In description of methods criteria of assessing weight status as normal weight, overweight and obese should be stated, as well as of the blood pressure.  In description of sample collection, examined biochemical parameters should be stated.

In the Key findings, a part of results is presented adequately for the protocol study. The strengths and weaknesses of the study are well described.

The number and choice of references is satisfactory.

--Thank you very much for the comments. We have added the classification and determination criteria of the relevant indicators in the methods section, as well as a detailed description of the biochemical indicators involved in the test (line 133-136; line 146-149; line 178-179).

Reviewer 3 Report

This is a scientific research.

The paper looks like preliminary findings (on-going study) instead of a proposal. Please clarify 

Participants were asked to take off all clothes except for underpants. How was the ethical procedure conducted in relationship to this?

Did the male physicians conducted examination on male students? Or they also checked opposite gender?

How was the issue of integrity tackled?

How was informed consent obtained?

Although this appear to be an ongoing study, discussion section should be provided and conclusion of what is new and how study contributed to new knowledge?

Require English proof read to improve paper.

Author Response

This is a scientific research.

The paper looks like preliminary findings (on-going study) instead of a proposal. Please clarify.

--Thank you very much for the comments. The present manuscript was a cohort profile for an ongoing cohort study, we have clarified the type of study in both titles and introductions (line 67-69).

Participants were asked to take off all clothes except for underpants. How was the ethical procedure conducted in relationship to this?

--Thank you very much for the comments. The ethical declaration for this study described in detail the examination situation: all examinations requiring the removal of outer clothing were performed in a completely closed room, and the project team would make sure that the doors, windows and curtains were closed before the start of each day's examination. During the examination, only one entrance and exit to the room is reserved and is covered by a screen to ensure that students' privacy is protected. This process was approved by the ethics committee and was strictly followed throughout the study. We have added these descriptions to method section (line 166-173).

Did the male physicians conducted examination on male students? Or they also checked opposite gender?

--Thank you very much for putting up this concern. Physical examinations including height, weight and Tanner stage request removing clothes, and were therefore conducted in private and closed rooms and by physicians with the same biological sex. Therefore, male physicians only conducted examinations for male students, while female physicians only conducted above examinations for female students. We also added these descriptions to method section (line 171-173).

How was the issue of integrity tackled?

--Thank you very much for putting up this concern. In the present cohort study, integrity issue was addressed through following practice: 1) study design: this study was designed with rigid scientific and ethical review before implementation, and was conducted strictly with the reviewed form of implementation protocol; 2) during data collection: research team provided technical guidance and participated in both data collection and quality control procedure. On each day of data collection, 5% of subjects would be randomly selected for a quality control recheck, all examinations would be repeated if the recheck fell to satisfy the implementation protocol (which did not happen in practice); 3) data handling: all data entry were conducted by two separate stuff to reduce processing error, all biological samples were strictly preserved according to the requirements and transported to a laboratory with testing qualification in Beijing for uniform testing after the completion of all field work. We added a separate part of quality control to address above issues (line 206-220).

How was informed consent obtained?

--Thank you very much for putting up this concern. Prior to participants recruitment, we held a parent meeting with students and their parents from targeted grade levels in each of the project school. The meetings were held by research team and school nurses, and introduced the objectives, specific contents and implementation methods of the cohort. Students and their parents were given adequate chance and time to ask questions. Thereafter, written informed consent forms were given to students and parents. They were told to bring the informed consent forms back home, to fully discuss it with student and family members, and to decide whether they would take part into this study. Those who agreed to participate in the study would submit an informed consent form signed by both student and the parent the next day. We have added these descriptions in method section (line 90-108).

Although this appear to be an ongoing study, discussion section should be provided and conclusion of what is new and how study contributed to new knowledge?

--Thank you very much for the comments. We divided the original “key findings” section into two parts, named primary results and discussion, to display what we found and how this study contributed to this field, respectively. We also added a conclusion part to summarize above information (line 245-248; line 266-271).

Require English proof read to improve paper.

--Thank you very much for the comments. We have carefully read through the manuscript and improved language.

Round 2

Reviewer 3 Report

The revision was made accordingly. The paper contributes to new knowledge in the scientific field.

Author Response

We do appreciated the comments from the reviewer.